# Analysis of the Passenger Flow Transfer Capacity of a Bus-Subway Transfer Hub in an Urban Multi-Mode Transportation Network

**Yuwei Yang** [1,2,3] **, Jun Chen** [1,2,3,*] **and Zexingjian Du** [1,2,3]

1 Jiangsu Key Laboratory of Urban ITS, Southeast University, Nanjing 211189, China;
   summer.yyuwei@gmail.com (Y.Y); duzexingjian@hikvision.com (Z.D.)
2 Jiangsu Province Collaborative Innovation Center of Modern Urban Traffic Technologies,
   Southeast University, Nanjing 211189, China
3 School of Transportation, Southeast University, Nanjing 211189, China
* Correspondence: chenjun@seu.edu.cn; Tel.: +86-025-5209-1279

**Abstract:** In the context of a rapid developing urban economy and the increasing number of motor vehicles, urban commuting transportation has witnessed a serious mismatch between the supply of and the demand for transportation network resources. In developing an urban multi-mode traffic network, using a urban traffic transfer hub to coordinate the transportation capabilities among different traffic networks is perceived to be highly effective for exploring the network transportation capacity of an entire transportation system, and improving travel efficiency and experiences for the public. Based on the super-network model, this paper develops a topological structure for a multi-mode traffic network, in which two typical combined travel modes are selected to establish the path impedance function for that network. Moreover, the multi-mode traffic allocation model and the solving algorithm are constructed in the research. The paper studies the impact of two types of factors related to the transfer capacity of passenger flows based on the regular traffic network of a bus-and-subway transfer hub using a sensitivity analysis of the transfer time and the associated penalty. The findings suggest that both changes in transfer walking time and the transfer penalty have no significant effect on the bus passenger flow.

**Keywords:** multi-mode traffic network; transfer hub; passenger flow transfer capacity

## 1. Introduction

In recent years, urban traffic modes have been transforming from a single-mode network into a multi-mode traffic network composed of a subway rail network, conventional bus network and road network, which have formed a modern urban traffic system. However, the lack of efficient connections among different traffic modes has led to serious local traffic network congestion. The concept of the urban multi-mode traffic network is not a simple superposition of different traffic networks. Instead, it collaboratively connects those networks through the traffic transfer hub, producing a whole that is greater than the sum of the parts. Pressures affecting transportation capacity could be transferred to more sufficient traffic networks through transfer hubs, so as to maximize the overall efficiency of the integrated traffic system.

The transportation transfer hub is an important link in a multi-mode traffic network. Its main advantageous feature would be the reasonable planning and organization that increase the competitiveness of combined travel modes and enable the public to enjoy higher efficiency and comfort. A multimodal transport network relies on the transport hub to promote cohesion of entire traffic networks and ensure there is efficient and orderly operation at all levels throughout the networks.

'Bus + Subway' is the most important combined transport mode in commuter travel, which facilitates a huge number of passenger flows transferring in urban multi-mode networks. However, due to large increases in urban populations, current transportation hubs cannot meet the demand for passenger flow transfers in multi-mode networks. To compensate for such a shortcoming, it is necessary to study the transportation hub from the perspective of a urban multi-modal transportation network structure. In particular, the combined travel mode enhances passenger flow transfer capability and emphasizes the role of the transportation hub in the entire multi-mode network.

This paper takes the multi-mode traffic network and transfer hub as the research object and constructs an impedance function of a multi-mode traffic network to establish a multi-mode traffic network topology. The research has explored the influence of a transfer hub on the transfer capacity on passenger flows of the multi-mode traffic network and has selected practical cases to analyze the sensitivity of the transportation hub.

The remainder of this paper is structured as follows: Section 2 presents a relevant literature review, followed by Section 3 demonstrating the topological structure for a multi-mode traffic network. Subsequently, Section 4 exhibits two types of path impedance functions of an urban multi-mode traffic network. Section 5 demonstrates the Multi-mode Traffic Allocation Model and Algorithm based on Stochastic User Equilibrium Model (SUE) and a case analysis. Section 6 analyzes the impact of the transfer walking time and transfer penalty through sensitivity analysis. Finally, the conclusion is provided.

## 2. Literature Review

### 2.1. Multi-Mode Traffic Network Mode Characteristics

Characteristics of travel, travelers and traffic patterns are the three main influential factors affecting individuals' choice of travel path. Among these factors, the traffic mode characteristics include cost, time, safety, comfort and accessibility [1–3]. He [4] indicated that the comfort, economic factor and time factor all affected a citizens' travel cost model. The Logit model improved the utility function by using Transcad to improve the Logit model's parameter calibration. The use of car travel and parking fees are the main two factors that ultimately influence the choice of how to travel. Li [5] has built a segment impedance function for a multi-mode traffic network that is associated with transfer penalties, travel times and travel costs.

### 2.2. Multi-Mode Transportation Network

The premise of traffic network allocation is to establish an urban traffic network topology. At present, a large number of scholars [5–9] have studied the network's topological structure and a corresponding mathematical model of a single traffic mode network, which mainly involves graph theory has emerged. This model views the network structure of each single urban traffic mode as a geometric network made up of different nodes and edges. However, there has been a lack of research on multiple-mode transportation and the connections between different traffic modes. As a result, the research focus has shifted to the study of multi-mode traffic network models. The super network model, GIS road network model, graph theory model and state transition network are the four main network building technologies for multi-mode traffic networks. The inner structure of an urban multi-mode traffic network is complex due to various transfer hub nodes and the transformation among traffic modes. For a better understanding of this structure, it is essential to study typical combined-travel modes and the distribution of passenger flows. For instance, Bayan et al. [10] have used the road network model to present all road network intersections and transfer points and have proposed a practical and feasible capacity assessment method for multi-mode traffic networks. Moreover, the research conducted by Li [4] has adopted a stochastic user equilibrium distribution model to complete urban traffic network allocation and demonstrated the connections between three sub-traffic networks based on the super network model.

To learn network operation status, the traffic network allocation model is essential. Public traffic network allocation model includes user equilibrium model (UE), stochastic user equilibrium model (SUE) and two-tier programming model. As indicated, people follow the Wardrop-first principle. After the urban traffic network, the equilibrium traffic theory assumes travel time of each passenger flow path is equal and minimal. Travel time of paths with no other road flows is longer than the shortest path [11–14]. However, such hypothesis is not valid in the actual scene. Travelers could not fully acquire traffic information of entire urban traffic networks through navigation software. They normally estimate their choices of travel path and choose the shortest route. A single traveler might not change the travel cost of their expected travel path. In this case, assuming travelers follow the first principle of Wardrop, the issue comes to the stochastic user equilibrium model (SUE) problem proposed by [15–17]. With the update of technology, Liu [18] uses a new generation technology to explore the short-term metro passenger flow prediction and distribute passenger flow.

Seen from the above, there have been certain research attempts to study the multi-mode traffic network model. Among the studies, the multi-mode traffic network model based on the super network stands out. It could reasonably express the spatial location characteristics of multi-mode traffic networks, reflect the correlation between different traffic modes and fully describe the travel process. However, current research is not accurate enough to express the transfer relationship in the super network. Thus, studying the impedance of virtual road segments is urgently needed. By comparison, SUE allocation model can well reflect the random behavior of travelers' choosing routes. To simulate the actual distribution of multimodal transport network, changes in travel factors like road impedance function should be studied. Subsequently, this paper will further the study of the multimodal transport network and the impedance function of transport hub node by using the super network to build the multimodal transport network topology. SUE model is adopted to establish the multimodal transport network traffic allocation in terms of the passenger transfer ability of transportation hubs.

### 2.3. Transportation Hub

At present, the generalized cost function of combined travel behaviors not only considers the behavioral factors of different transportation modes, but also the factors of transfer facilities and transfer utility among those modes. Scholars have examined six influential factors of combined travels: personal attribute of travelers, travel purpose, traveler's mood, transfer convenience, transfer fees and the flexibility of transferring hub. Other scholars also indicate factors such as the transfer walking time and waiting time [19–22].

Chowdhury and Ceder [23] believe that the transfer penalty is strongly related to the internal corridor structure of the hub. If the internal transfer of the transport hub is smooth enough, travelers will choose combined travel methods. Chowdhury [24] found that passengers were intolerant with transfer delay. Therefore, minimizing transfer walking time and waiting time are expected to improve the probability of choosing combined travels. Comparing with single travel mode, transfer travel causes psychological resistance among travelers. Researchers often quantify transfer punishment in their study of the transfer penalty. For example, Yoo [25] quantified the transfer penalty into bus traveling time and found that the transfer penalty values perceived by passengers on different routes were different in South Korea.

### 2.4. Multi-Mode Network Segment Impedance Function

Section impedance is the key to the traffic network allocation. Most scholars focus on the composition of road impedance in previous studies in the field of multi-mode networks. There are abundant research findings around the impedance function of single traffic mode. For instance, BPR function is a car road impedance function often used at home and abroad. However, such function is built on the highway data in the United States, which is not well compatible with domestic urban roads in China [26,27]. Subsequently, many scholars attempt to optimize BPR function to build the section impedance function under domestic road conditions. Gai [28] studied the selection of the basic

capacity of road sections and constructed the section impedance function model based on the road toll. Wang et al. [29] established the road impedance function by taking time, cost, comfort and other factors and the generalized impedance concept. Moreover, Si et al. [5] build the section impedance function of urban multi-mode traffic networks by engaging the influence of different travel modes in the road network. Li et al. [30] perceive that the variables of road impedance function include transfer time, travel time, platform waiting time and walking time. By limiting the number of transfer times, the choosing behavior of passengers is studied through transfer punishment. Sun [31] indicated that the generalized travel path cost of subway passengers was the weighted average sum of travel time, transfer time and the passenger flow density index. In addition, Lin et al. [32] illustrated that the main factors affecting passengers' route selection in urban rail transfer networks include travel time, number of transfers, transfer time, and the constructed impedance function by punishing transfer time.

Based on the above, there is a lack of research on the road impedance of the integrated transportation system composed of cars, conventional buses and subways. This paper is limited to the superposition of section impedances of different traffic modes. The study on the impedance of transfer hubs is limited.

## 3. Establishment of Topological Structure

### 3.1. Travel Characteristics of Urban Multi-Mode Traffic Network

The paper studied the slow traffic mode as daily transportation, involving cars, regular buses and rail subways. Regardless of the city size, travelers are now able to choose from a variety of transportation tools, which can be categorized into the combined mode of transportation and the single mode of transportation.

The modes of transportation studied in this paper are cars, regular buses and subways. The combined travel modes studied in this paper include "car + subway" and "regular bus + subway". The travel characteristics are as follows:

### 3.1.1. Analysis of Single Trip Characteristics

When travelers prioritize the directness of trip, they prefer to choose the combination of transportation without transfers. The car is considered as the most popular transportation tool. The single travel mode studied in this paper includes car travel mode, regular bus travel mode and subway travel mode. However, in many regions, it is difficult for travelers to directly reach the destination only by a single mode of travel. For example, if all travelers choose to travel by car, there will be inevitable road traffic congestions and insufficient use of other public transportation resources. So, there is the need for coordinating other modes of transportation and combined trips.

### 3.1.2. Analysis of Combined Travel Characteristics

First, the travel mode of car transferring to subway is the combination mode most suitable for people living in the suburbs and working in the downtown to avoid traffic congestions in rushing hours. The travel mode can reduce travel time, costs, especially in recent years, along with the rise of sharing cars, making this combination mode prevailing. Thus, the proportion of car-to-subway mode has significantly improved in all the travel modes. At present, China is actively building park-and-ride parking lots, and by introducing the annual pass, seasonal pass and other preferential parking policies to encourage travelers to choose the combined travel mode.

Second, the mode of regular bus transferring to subway is introduced. Bus has always been the top choice of most Chinese on their selection of transportation. Travelers will choose regular bus + subway or subway + regular bus as combined travel mode. This travel mode charges low fares. It travels stably and punctually through fixed bus network and rail network, which are not affected by road traffic conditions. Regular bus and subway services are stable while the travel time is reliably reduced. Additionally, the network covers wide areas, making up for the shortage of direct rail subways.

The urban multi-mode traffic network consists of multiple traffic modes. Aforementioned modes present the analysis of the characteristics of current multimodal transport network. Considering that the super network is advantageous in describing the operation condition, the correlation among different traffic networks and the characteristics of transfer hubs, this study will apply this model to construct the urban multi-mode traffic network topology.

Though urban multi-mode traffic system involves various traffic modes, this paper only studies slow traffic as cars, regular buses and subways. In order to construct the topology of multi-mode traffic network, it is necessary to replace the transfer hub node that connects two traffic modes with the virtual transfer section, which include features such as transfer cost, transfer walking time and transfer penalty. Before reaching the nodes of the multi-mode traffic network from the starting point of travel, travelers need to take slow traffics like walking or cycling, and leave before the end of travel. Therefore, the multi-mode transportation network involves the connecting networks represented by the slow traffic. Assuming that the network only has attributes such as travel time, in this paper, the abstract topology of road network structure section is set based on travelers' morning rush hour commutes:

For the car road network, the urban branch road network near the travel OD point abstracts the road intersections as nodes and the urban main commuter road and the expressway into the topological structure by taking the main road intersections as nodes.

The conventional bus network abstracts the main bus line site as a node topology. The subway network takes the subway station as a node and abstracts the subway line into a topology.

To sum up, the super-network is well used to construct urban multi-mode traffic network. To specify, the traffic network where various traffic modes are located is a relatively independent sub-traffic network. The nodes of the sub-traffic network represent slow-moving nodes, intersections, transfer hubs, bus stations, subway stations and so on. The line segments between the two nodes represents sections. Sub-traffic networks have different operating characteristics and section impedances. Each sub-transportation network is connected through the transfer hub. In the super-network topology, dotted lines are used to replace the transfer hub node. Furthermore, the virtual path of the transfer hub has many impedance attributes that affect the travel cost, such as walking time, waiting time and transfer cost. The core construction steps of multi-mode traffic network topology include the following.

Establish a network of different sub-modes of transportation and connectivity. Based on the characteristics of traffic mode and operation as well as the actual spatial location of the urban network, the sub-network of a single traffic mode is constructed by the method of graph theory.

Based on the transfer hub of multi-mode traffic networks, the transfer correlation among sub-networks is established. By setting the transfer hub as a virtual road section, the physical nodes at the same position at different levels are connected to represent the transfer relationship. The topological structure of urban multi-mode traffic network is constructed by superimposing each sub-traffic network regularly.

## 3.2. Network Construction Method of Net and Offline Network

As shown in Figure 1, the connecting network section is transferred by slow traffic or turns out the sub-traffic network section. The connecting network section is represented by dotted line. When the connecting network section is connected to the sub-traffic network of cars, it only has the section passage time attribute. When connected to the public transport network, there are other attributes such as bus fares. Generally, there is no monetary cost except the time cost in the connect network.

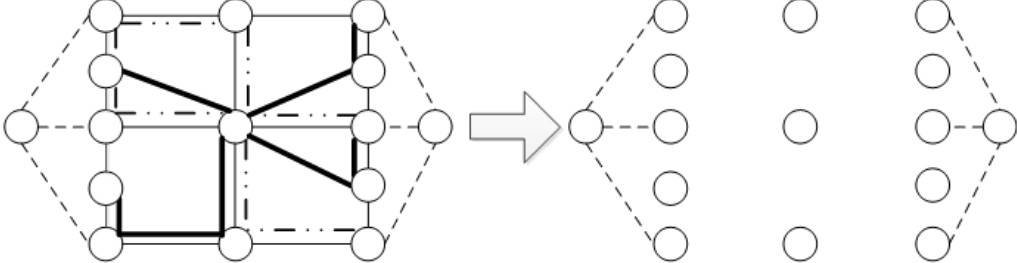

**Figure 1.** Schematic Diagram of Connecting Network Subnetwork.

*3.3. Subnetwork Construction Method of Cars*

As shown in Figure 2, the car subnetwork is composed of nodes representing intersections and parking lots. The line segments represent road sections. The transfer node is in the parking lot. The actual physical nodes and sections are represented in the network. Among them, each solid line segment represents a variety of attributes, such as section length, section capacity, section flow, section passage time, section passage cost, etc.

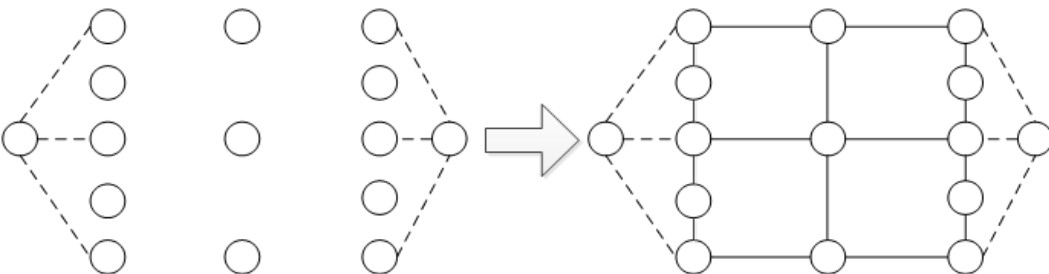

**Figure 2.** Schematic Diagram of Car Subnetwork.

*3.4. Regular Construction Method of Bus Subnetwork*

As shown in Figure 3, the regular bus subnetwork is an independent traffic subnetwork, while the urban bus network is composed of stations, sections and routes, and these three elements are interrelated. For one thing, at transfer stations, different lines can switch between each other. On the other hand, the same station and the same section can contain multiple bus routes, each of which has its own operational characteristics. The solid line of bus network can represent a variety of attributes, such as section length, section travel time, number of passengers, etc. In this paper, the regular bus network topology model is established by using extension technology.

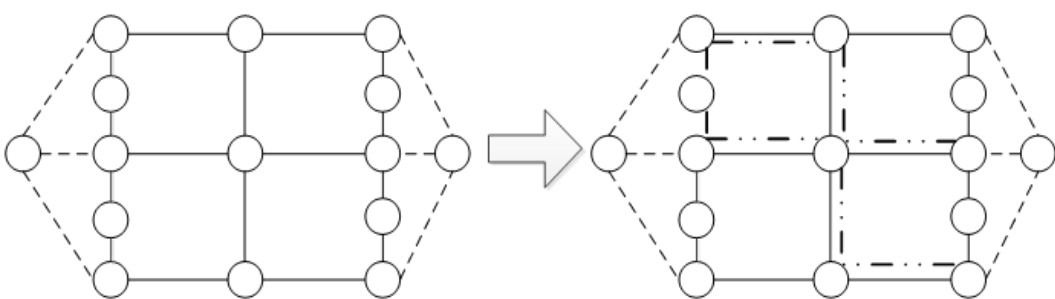

**Figure 3.** Schematic Diagram of Regular Bus Subnetwork.

*3.5. Subway Subnetwork Construction Method*

As shown in Figure 4, the subway subnetwork is also an independent traffic subnetwork, which is composed of stations, sections, and lines. Unlike a regular bus network, there is usually only one

line running on the same section, and at transfer points, different lines can be transferred to each other. The solid segment of the subway network can represent a variety of attributes, such as the length of the segment, the number of passengers carried, the running time and so on.

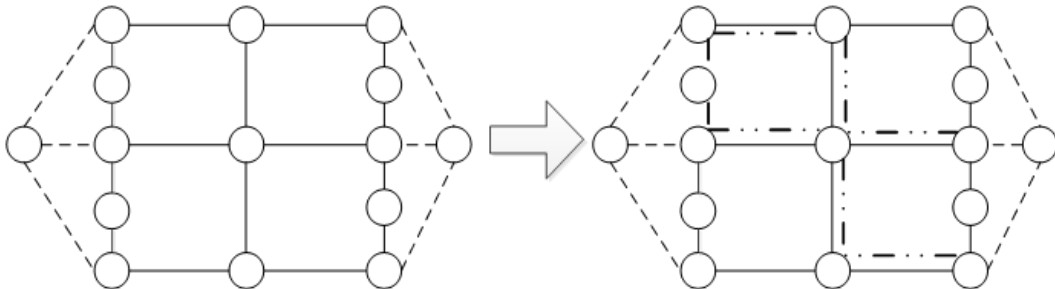

**Figure 4.** Schematic Diagram of Subway Subnetwork.

After constructing each sub-network of multi-mode transportation network, corresponding to the transfer stations of each network, a multi-modal transportation network hub is added to transfer virtual road sections. The types of interchange hubs include regular bus, subway track and car, and subway track. Transferring virtual road segment attributes include parking fees, transfer waiting time, transfer walking time, bus fare, transfer penalty and others.

The constructed multi-mode traffic network topology is shown in following Figure 5.

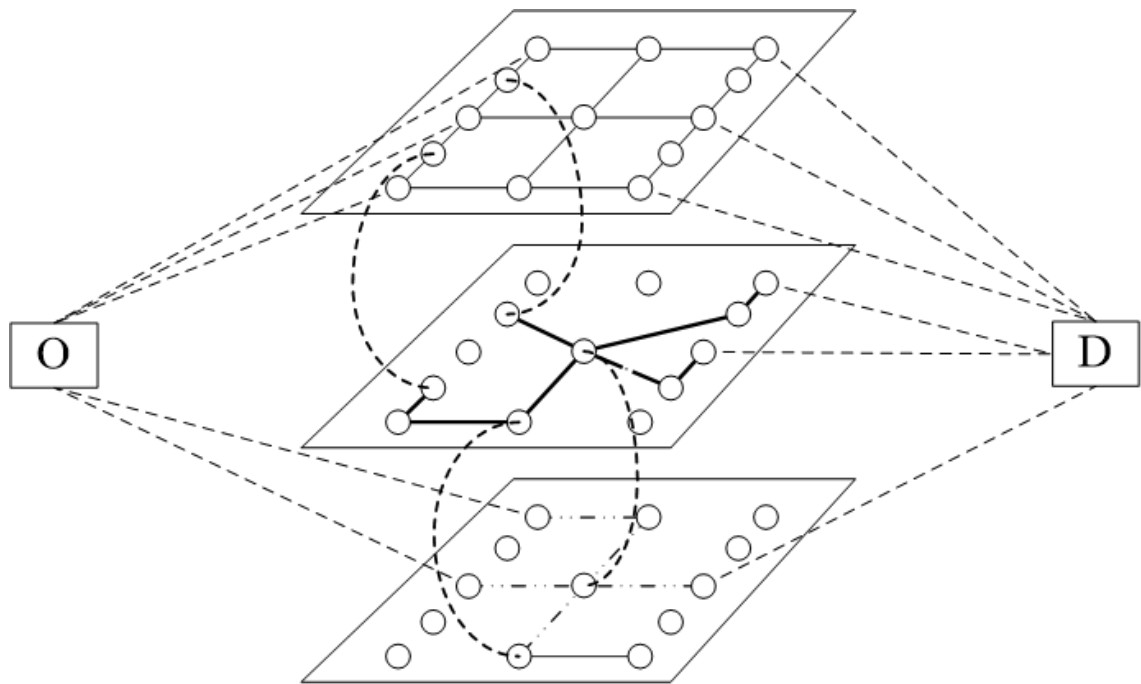

**Figure 5.** Multi-mode Traffic Network Topology.

## 4. Urban Multimode Traffic Network Impedance Function

The urban traffic road impedance is the main factor that influences passenger choosing travel segments and routes. Travelers' travel costs consist of transfer fees, travel expenses, and walking time. The transfer fee includes transfer penalty, transfer walking time and transfer waiting time. The travel expenses mainly include the travel time and transportation charges. In general, factors affecting passenger travel choices are passenger class, transfer time, travel time, and number of transfers.

The multi-mode traffic network section impedance function in this paper is based on the traffic travel in large and medium-sized cities during the morning rush hour commute, through the analysis

and comparison of the influencing factors of passengers' travel choice, and the determination of different research objects and applicable conditions. As a whole, the section impedance function of different traffic modes in this paper is composed of three parts: Driving time, travel cost and comfort loss, which are specified as Equation (1):

$$I_a^n = T_a^n + P_a^n + D_a^n \tag{1}$$

where, $n$ denotes the mode of transportation ($c$ denotes car, $m$ denotes subway track, $b$ denotes regular bus), $a$ denotes section, $I$ denotes section impedance, $T$ denotes time cost, $P$ denotes monetary cost, and $D$ denotes comfort loss. The time cost includes driving time and waiting time; monetary costs include fuel costs, line fares and parking costs. This paper argues that travelers' sense of comfort depends on the mode of transportation and travel time.

### 4.1. Connection Network Section Impedance Function

The impedance function of the connecting network segment includes the time cost and the fare transferred by the destination vehicle. For the connecting network of the car network, the fare cost is not included while the comfort loss is not considered. This research assumes that the connecting network adopts the mode of walking. The expression is as Equations (2)–(4):

$$T_a^u = t_a^{u0} + t_a^{u1}, a \in A^u \tag{2}$$

$$P_a^u = \eta \times \cos t^u \tag{3}$$

$$D_a^u = 0 \tag{4}$$

where, $A^u$ represents the connecting network, $t_a^{u0}$ is the walking time of travelers on the $a$ connecting network, $t_a^{u1}$ is the average waiting time, $\cos t^u$ means fares for the metro network or regular bus network, $\eta$ is the dimensional conversion parameter, that is, the cost - time conversion coefficient.

The connecting network is similar to the connecting network. However, it only considers walking time. Travel cost and comfort loss of the section are all calculated as zero. The expression is as Equations (5)–(7):

$$T_a^d = t_a^{d0}, a \in A^d \tag{5}$$

$$P_a^d = 0 \tag{6}$$

$$D_a^d = 0 \tag{7}$$

where, $A^d$ denotes the connecting network, $t_a^{d0}$ denotes the walking time of travelers on the $a$ connecting network.

### 4.2. Car Network

#### 4.2.1. Travel Time

As a private travel tool, car is featured by highest comfort. The travel time of the car road section is limited by the traffic flow and the road network capacity. In the paper, traffic delay at road intersections is not considered. The passage time can be given by BPR function as Equation (8):

$$T_a^c = t_a^{c0}[1 + \alpha(\frac{v_a}{\lambda^c C_a^c})^\beta], a \in A^c \tag{8}$$

where, $A^c$ denotes the collection of sections for the car network; $t_a^{c0}$ denotes the ideal driving time of cars on the $a$ road; $v^a$ refers to the pedestrian flow (person-times /h) on the $a$ road; $C_a^c$ represents the volume of cars on the $a$ road (pcu/h); $\lambda$ represents the conversion coefficient (person-time/pcu) of

passenger flow into traffic flow on the car network, that is the average number of passengers carried by the car; $\alpha$ and $\beta$ are the undetermined coefficients respectively.

### 4.2.2. Monetary Fees

The monetary cost of car traveling in the network is mainly fuel cost. The calculation formula is as Equation (9):

$$P_a^c = \eta \times \rho \times x_a \tag{9}$$

where, $x_a$ means the length of $a$ road (km); $\rho$ is fuel cost (yuan /km), $\eta$ is the dimensional conversion parameter, that is, the cost-time conversion coefficient.

### 4.2.3. Comfort Loss

The comfort loss function of a car is related to its travel time on the road. The specific formula is as Equation (10):

$$D_a^c = \omega \times s_a^c \times T_a^c \tag{10}$$

where, $T_a^c$ represents the driving time of the car on the $a$ road; $s_a^c$ is the loss of comfort of the traveler per unit time in the car (/h); $\omega$ is comfort loss-time conversion factor.

Some parameters are not marked because this paper only studies commuters in the morning rush hour. As the same user type, travelers share consensus on the equivalence relationship between comfort loss and time.

### *4.3. Regular Bus Network*

Conventional buses are the main means of transportation because of low travel costs. The comfort and reliability are worse than those of cars. However, conventional buses are well performed if they are driven on special bus lanes. Besides, the travel time is relatively fixed. Considering those characteristics, the routine bus network section impedance function in this paper is composed as follow:

### 4.3.1. Travel Time

Assuming that the regular bus travels on the bus lane during the rush hour, the travel time between the bus stops can be regarded as a constant, that is Equation (11):

$$T_a^b = t_a^{b0}, a \in A^b \tag{11}$$

where, $A^b$ represents the assembly of the regular bus networks, $t_a^{b0}$ represents the driving time of regular buses on the $a$ road.

### 4.3.2. Monetary Cost

Regular bus does not charge by mileage but adopts fixed pricing strategy of line fare. The fare cost could not be shared among all bus networks. The multi-mode traffic network topology to be built in this paper considers that passengers pay for buses after boarding, thus bus fares are transferred to the connecting network or transfer network. As Equation (12):

$$P_a^b = 0 \tag{12}$$

### 4.3.3. Comfort Loss

The comfort loss of regular bus network is affected by factors more than travel time, and related to the degree of congestion. The corresponding calculation formula of comfort loss of bus section is as Equation (13):

$$D_a^b = w \times s_a^b \times T_a^b + \left( \frac{v_a A}{BC_a^b A} \right) \tag{13}$$

where, $T_a^b$ represents the driving time of regular buses in the *a* road; $s_a^b$ is the comfort loss per unit time (/h) of travelers when taking bus; $\omega$ is the dimensional transformation coefficient, that is, the comfort loss-time conversion factor; *B* represents the designed passenger capacity (person-time/vehicle) of the bus; $C_a^b$ represents the capacity of regular bus in the *a* road (pcu/h). *A* represents vehicle conversion factor, according to the Highway Capacity Manual. *A* is usually valued by 2.

*4.4. The Subway Network*

Subway is a long-distance public travel mode with large traffic volumes. The cost is very similar to that of regular buses. Since the subway operates in an independent subway network, there are no interferences among the vehicles. The travel time is not affected by network passenger traffic congestion. Considering these characteristics, each part of the section impedance function of subway rail network in this paper is as follow:

4.4.1. Travel Time

Since the subway operates in an independent subway network, the travel time of the section is the same as that of the regular bus network, that is Equation (14):

$$T_a^m = t_a^{m0}, a \in A^m \tag{14}$$

where, $A^m$ means the collection of the rail and subway network, $t_a^{m0}$ represents the driving time on the *a* road section.

4.4.2. Monetary Cost

Like conventional bus, subway is a public transportation service mode, of which the monetary cost expression can also be modeled like bus network, that is Equation (15):

$$D_a^m = \omega \times s_a^m \times T_a^m + \left( \frac{v_a}{EC_a^m} \right) \tag{15}$$

where, $T_a^m$ represents the driving time of regular buses in the *a* road; $s_a^m$ is the comfort loss per unit time (/h) of travelers when taking bus; $\omega$ is the dimensional transformation coefficient, that is comfort loss-time conversion factor; *E* represents the designed passenger capacity (person-time/vehicle) of subway cars; $C_a^m$ means the traffic volume of subway track in the *a* road (pcu/h).

*4.5. Internal Transfer Section Impedance Function*

The transfer section in this section is internal within the mode. The impedance function of this section also includes three parts: time cost, transfer cost and transfer penalty cost.

4.5.1. Time Fee

Time fee of the transfer section includes the walking time and transfer waiting time. As Equation (16):

$$T_a^i = t_a^{i0} + t_a^{i1}, a \in A^i \tag{16}$$

4.5.2. Monetary Cost

That is Equation (17):

$$P_a^i = \eta \times \cos t^i \tag{17}$$

### 4.5.3. Transfer Penalty Cost

Due to the switch between vehicles in traffic network, travelers feel transfer penalty as extra inconvenience. This paper converts this feeling into the impedance function of the road section during the traffic time. As Equation (18):

$$S_a^i = penalty_a^i \qquad (18)$$

where, $A^i$ represents the collection of transfer sections; $t_a^{i0}$ is the walking time on the $a$ transfer route; $t_a^{i1}$ is the average waiting time of travelers on the $a$ transfer road; $\cos t^i$ is the cost of transfer to a subway or regular bus network. This paper assumes that the car network does not involve internal vehicle transfer behaviors. Thus, the car network values zero; $penalty^i$ is the equivalent of the transfer time in the car time (h).

The transfer hub is an important link in the multi-mode traffic network. Travelers choose the combined travel mode and transfer through the hub. In this paper, transfer cost, transfer walking time, waiting time and transfer penalty are proposed to describe the influence of multi-mode transportation transfer hub on travelers' choosing combined travel mode. Besides, this paper highlights the transfer punishment, which is the extra psychological burden caused by the transfer and quantifies it as equivalent in the car time.

In order to build a multi-mode traffic network topology model based on super network, the traffic transfer hub is assumed as a virtual transfer path. The impedance function is constructed as Equations (19) and (20):

$$H_a^g = t_a^{g0} + t_a^{g1} + \eta \times \cos t^g + penalty_a^g, a \in A^g \qquad (19)$$

$$H_a^d = t_a^{d0} + t_a^{d1} + \eta \times (\cos t^d + park^d), a \in A^d \qquad (20)$$

Two kinds of multi-mode transport network transfer hubs studied in this paper are the conventional bus transfer hub (Conventional bus transfer hub and metro transfer hub) and the car transfer hub (Park and Ride (P + R) transfer hub).

Where, $A^g$ means the collection of virtual transfer sections of conventional bus and subway; $t_a^{g0}$ is the walking time on the $a$ virtual transfer section of conventional bus and subway transfer hub; $t_a^{g1}$ means the average waiting time on the $a$ virtual transfer section of conventional bus and subway transfer hub; $\cos t^g$ is the fare for the transfer network; $A^d$ means the collection of virtual transfer section of car to subway; $t_a^{d0}$ is the walking time on the $a$ virtual transfer section of P + R transfer hub; $t_a^{d1}$ means the average waiting time on the $a$ virtual transfer section of conventional P + R transfer hub; $\cos t^d$ is the transfer fare to the rail network; $park^d$ is the parking fee.

The path impedance function of urban multi-mode traffic network is extended based on the section impedance function of urban multi-mode traffic network. If passengers fully understand the traffic condition of urban multi-mode traffic network, during the trip, they will estimate the traffic impedances of all possible routes in each OD pair of a trip and make comparison and take the route with the lowest impedances. According to the section impedance function of different sub-traffic networks constructed in the previous section, the route impedance function of the travel chain selected by travelers can be obtained.

As the Figure 6 presents, the red line $a$ shows the complete travel path of a traveler, who chooses the combined travel mode of car and subway. The $a$ path impedance function is Equation (21):

$$I_a = T_{a1}^u + P_{a1}^u + T_{a2}^c + P_{a2}^c + D_{a2}^c + H_{a3}^g + T_{a4}^m + D_{a4}^m + T_{a5}^d, a_1, a_2, a_3, a_4, a_5 \in a \qquad (21)$$

where, $I_a$ is the equivalent time of the traveler's path $a$; $T_{a1}^u$ is the time cost in the $a_1$ connecting network; $P_{a1}^u$ means monetary fees for travelers on the $a_1$ connecting network; $T_{a2}^c$ is the travel time in the car network; $P_{a2}^c$ is the monetary costs for travelers in the car network including fuel consumption; $D_{a2}^c$ means the comfort loss of the traveler in the car network; $H_{a3}^g$ means the impedance function of travelers at P + R transfer hub; $T_{a4}^m$ means travelers' time expense in the subway track network;

$D_{a4}^m$ means travelers' comfort loss in the subway network; $T_{a5}^d$ means the travelers' time cost in the $a_5$ connecting network section.

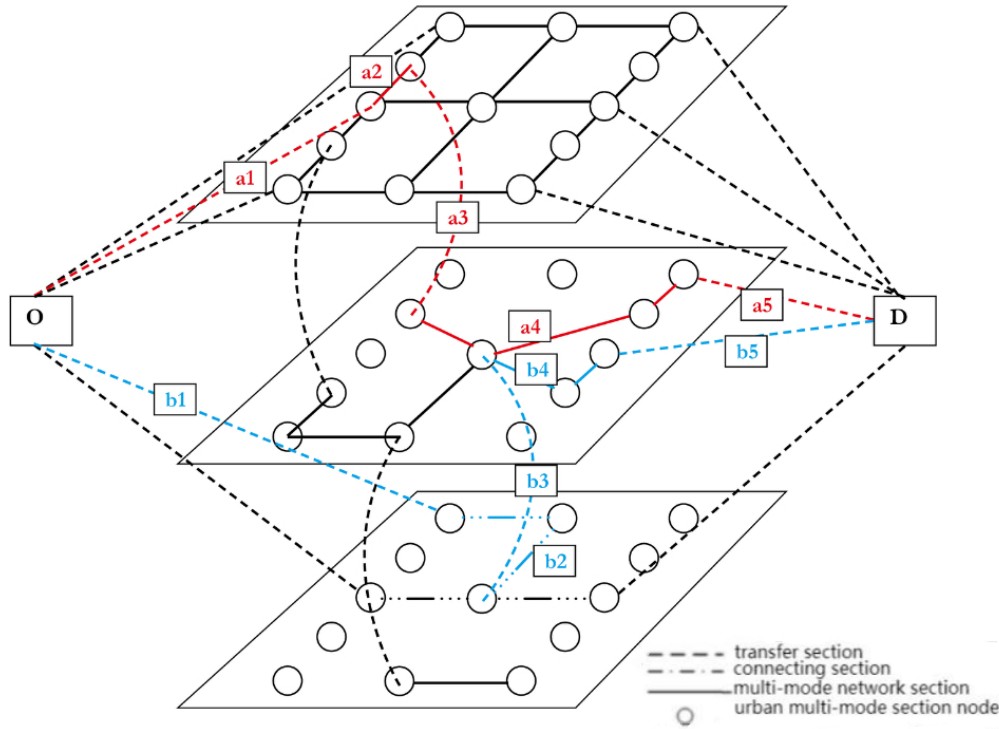

**Figure 6.** Car + Subway Combined Travel Mode and Bus+ Subway Travel Chain.

In the Figure 6, the $b$ blue line segment shows the complete travel path of a traveler who takes the combined travel mode of conventional bus and subway. The $b$ path impedance function is as Equation (22):

$$I_b = T_{b1}^u + P_{b1}^u + T_{b2}^b + D_{b2}^b + H_{b3}^d + T_{b4}^m + D_{b4}^m + T_{b5}^d, b_1, b_2, b_3, b_4, b_5 \in b \qquad (22)$$

where, $I_b$ is the equivalent time of the traveler's path $b$; $T_{b1}^u$ is the time cost in the $b_1$ connecting network; $P_{b1}^u$ means monetary fees on the $b_1$ connecting network including bus fares; $T_{b2}^b$ is the time cost in the bus network; $D_{b2}^b$ means the comfort loss of traveler in the bus network; $H_{b3}^g$ means the impedance function of travelers at P + R transfer hub; $T_{b4}^m$ means the time cost in the subway network; $D_{b4}^m$ means the comfort loss of travelers in subway network; $T_{b5}^d$ means the time cost in the $b_5$ connecting network section.

## 5. Multi-Mode Traffic Allocation Model and Algorithm Based on SUE

Combined with the research of the multimodal transport network road impedance function, the analysis of multimodal transport networks, traffic network requirements and the road impedance function, and the relationship among the transferring hub connecting sections, based on the stochastic user equilibrium assignment model built on the demand flow conservation constraint conditions, the research has established a multi-mode network distribution model targeting at minimizing passenger traffic impedance based on the weighted average method (MSWA) in a row, and developed a multimodal distribution model algorithm.

In the context where urban traffic information is not transparent, the distribution of multimodal transport network traffic of a city is the result of travelers' choices. People always want to choose the path of the generalized cost minimum. However, as to choose a single traffic network with the increased number of commuters, the transport network will be more crowded, while the generalized cost will

increase, thus travelers can transfer the path to other traffic network layer and reach the dynamic equilibrium of multimodal transport network. The paper adopts the Stochastic User Equilibrium to describe the distribution Equilibrium status of urban multi-mode traffic networks. Considering Fisk model [33], a balanced distribution model of urban multi-mode traffic networks based on super network is proposed. When the urban multi-mode traffic network is in a balanced status, the path flow among all selected OD pairs should meet the following balance conditions, as Equations (23)–(26):

$$\min Z(f) = \frac{1}{\theta} \sum_w \sum_k f_k^w \ln f_k^w + \sum_a \int_0^{x_a} I_a(w) dw \tag{23}$$

$$s.t. \ \sum_k f_k^w = q^w, \forall k \in K^w, w \in W \tag{24}$$

$$f_k^w \geq 0, \forall k \in K^w, w \in W \tag{25}$$

$$x_a = \sum_w \sum_k f_k^w \delta_{a,k}^w, \forall a \in A \tag{26}$$

where, $f_k^w$ is the flow rate in the $k$ path between $w$; $q^w$ is the total travel volume between OD pairs in the $W$. $\theta$ represents to what extent passengers understand on the traffic information of the network. The larger the value is, the more comprehensive the traffic information is, and the less random the network will be. When the value is infinite, it represents the specific user equilibrium network model (UE model). $\delta_{ak}^w$ is the correlation between road section and path; $I_a$ is the generalized cost over radian $a$; $W$ represents the set of OD pairs in the entire traffic network; $K^W$ represents the collection of traffic paths in the entire traffic network.

### 5.1. Solution Algorithm

This paper used the continuous weighted average method [34] (MSWA algorithm) to solve the super network SUE model. Such algorithm allows a faster convergence speed than traditional sequential average algorithm (MSA algorithm) and a easier approach to the optimal. The procedures are as follow:

Step 1: Initialize. Assume an initial solution $x^0$ and make $y^0 = F(x^0)$. Number of iterations equals to 0. Set up the convergence parameter $\varepsilon$;

Step 2: Calculate $\|y^0 - x^0\|$, if $\|y^0 - x^0\| < \varepsilon$, and then stop calculating output $y^0$. Otherwise, it moves to the next step;

Step 3: Make $\chi^n = \frac{n^d}{1^d + 2^d + ... + n^d}$ where d is the correction parameter. When d is larger, the weight of the initial iteration value is smaller, and the convergence rate of the algorithm is faster, then $x^{n+1} = x^n + \chi^n(y^n - x^n)$, $y^{n+1} = F(x^{n+1})$;

Step 4: Iteration. Make n = n + 1, go to step 2, until the loop is complete.

Based on the MSWA algorithm, this paper solves the urban multi-mode traffic network allocation model by the following steps:

Step 1: Make the iteration number n = 0, and the flow matrix of each section of the urban multi-mode traffic network $x^{(0)} = 0$;

Step 2: Put the section flow into the generalized cost function, and calculate the cost of each section through the calculation formula $I_a^n$; Depth-First Search method (DFS) was used to determine OD pairs of all paths between $W$ [35] (DFS); Calculate the cost of each path and use the shortest path method to obtain the shortest path and the cost $I_{f,\min}$;

Step 3: Use Logit model to load the traffic flow of urban multi-mode traffic network path, and obtain the traffic flow of auxiliary sections $y^{(0)}$;

Step 4: MSWA method is used to obtain the section flow matrix of multi-mode traffic network in each city. Make d = 1. The formula is as (27) and (28):

$$x^{n+1} = x^n + \chi^n(y^n - x^n) \tag{27}$$

$$\chi^n = \frac{n^d}{1^d + 2^d + \ldots + n^d} \tag{28}$$

Step 5: Convergence test. If $\sqrt{\sum \left(x^{(n+1)} - x^{(n)}\right)^2} \times \left(\sum x^{(n)}\right)^{-1} \leq \varepsilon$, then $x^{(n+1)}$ is the final section flow solution, the calculation ends; Otherwise, set n = $n$ + 1 and return to Step 2.

### 5.2. Case Study

This section will take the local multi-mode traffic network in a certain area of Nanjing city as the evaluated network structure, analyze the morning peak commuting scene by using the transport capacity coordination analysis, process of urban multi-mode traffic network, and calculate the distribution of urban multi-mode traffic network and the transport capacity coordination evaluation index of multi-mode traffic networks.

### 5.2.1. Identify Research Objects

This example shows a commuter travel scene in the morning rush hour. From the suburb of Nanjing city, 7000 passengers are moving towards the city center namely Xinjiekou, in Xuanwu District. Figure 7 shows the scope of urban traffic networks for commuters during morning rush hours according to Baidu Map.

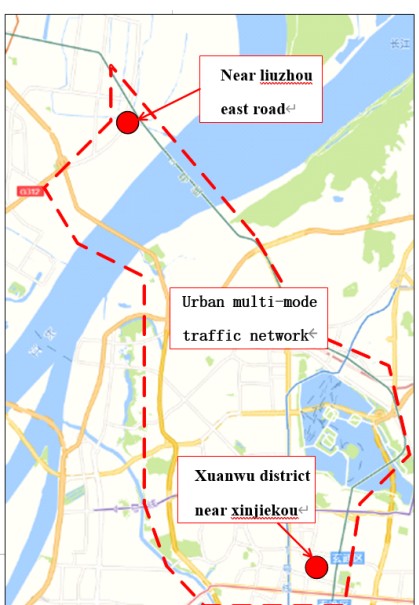

**Figure 7.** The Map Urban Multi-mode Traffic Network Coverage accessed from Baidu Map.

According to Baidu Map, the single network traffic path between OD points is as follow:

In the car network, travelers pass the Yangan Road and the branch road near Liuzhou East Road, cross the river through Jiangshan Road and Nanjing Yangtze River Bridge, and pass the expressway to the secondary road and the branch road near Xinjiekou, and finally reach the destination through the secondary trunk road and the branch road.

In the conventional bus network, travelers can walk to the bus stop and choose either bus No.532 or bus No.669 Inner Ring with transfer to bus No.532, and finally arrive at the bus stop near the destination and walk there.

In the subway network, travelers can walk to Liuzhou East Road subway station to take line 3 train. In the halfway, they can choose Nanjing station to transfer to subway line 1 to reach the subway station near the destination and walk there.

For the urban road transportation network, based on the road selection criteria, the study selected the urban traffic secondary trunk road near the OD point, the branch road network, the traffic commuting corridor trunk road and expressway as the basic structure of the road network. For the urban public transportation network, the bus station is selected as the regular bus network node, and the subway station as the subway network node. The research and will study the P + R interchange hub and the conventional bus and subway interchange hub. Following figure shows the diagram of a multi-mode traffic network.

Transportation facilities such as bus lines, subway lines and transfer stations in the urban road network involved in the study are shown in Table 1.

**Table 1.** Research scope network content table.

| The Network Level | Facility Category | Name of Traffic Facilities |
| --- | --- | --- |
| Urban road network | Secondary trunk road and branch road Arterial road | Liuzhou east road subway station near branch road A branch road near hongwu north road Jiangshan road, nanjing Yangtze river bridge, etc. |
| | Expressway | Daqiao north road, inner ring west road, huju north road, inner ring west line, etc. |
| Regular bus network | — | Bus No.532 Bus No. 669 |
| Subway network | — | Subway line 1 Subway line 3 |
| P + R transfer hub | — | Liuzhou east road subway station |
| Bus + Subway transfer hub | — | Venice watertown station |

### 5.2.2. Build the Topological Structure of Urban Network

Based on the super network model, the OD demand point connecting section, the car road network, the conventional bus network and the subway track network in the research scope are abstracted into a network topology structure. Various transfer hubs are converted into virtual transfer road sections.

Within the research scope, there are 11 connecting sections, more than 100 sections on the car road network, more than 20 regular bus sections, more than 20 railway sections, and 2 urban multi-mode traffic transfer hubs. According to the principle of segment abstract topology, the elements of urban multi-mode traffic network are constructed. The results are shown in Table 2.

**Table 2.** Table of topological sections of urban traffic network.

| The Network Level | Number of Road Sections (One) |
| --- | --- |
| Connection network | 8 |
| Car road network | 40 |
| Regular bus network (including internetwork transfer) | 7 |
| Subway network (including internetwork transfer) | 9 |
| P + R transfers virtual paths | 1 |
| Bus + subway transfer virtual path | 1 |

The topological structure was numbered with road elements, and finally the topological structure of each network level and urban multi-mode network was obtained. See Figure 8.

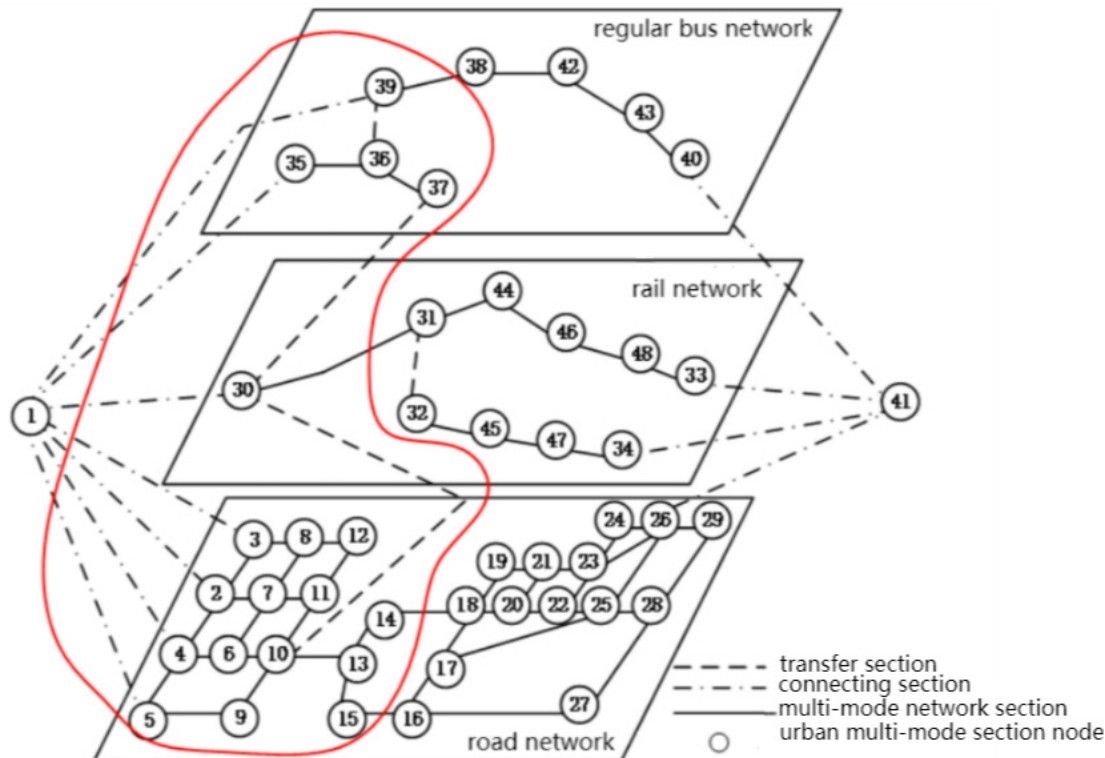

**Figure 8.** Urban Multi-mode Network Topology.

### 5.2.3. Urban Multimode Network Segment Impedance

The impedance of 68 urban traffic sections is calibrated. Partial parameters of urban multi-mode traffic network section impedance are obtained through references and Internet access. Where, $\alpha = 1.19$, $\beta = 3.09$, $\lambda_c = 1.4$. The per capita income of Pukou district in Nanjing was 39,906 Yuan in 2016, therefore $\eta = 251 \times 480/39906 = 3.02$ min/Yuan; $p = 0.66$, $w = 0.5$, $s_a^c = 0.1$, $S_a^b = 0.8$, $s_a^m = 0.5$. The capacity and speed of the road network are shown in the Table 3, the capacity of bus network is $BC_a^b = 1200 person/h$ and the rail capacity of the rail network is $EC_a^m = 5000 person/h$. Waiting time of both bus and subway is 3 min. Transfer time of P + R transfer hub is 10 min. Parking fee is 16 Yuan. Bus-and-subway transfer time is 7 min. The punishment time of internal transfer of bus, subway or bus-and- subway is 10 min.

**Table 3.** Section basic information table.

| Road | Traffic Volume (pcu/h) | Theoretical Velocity (km/h) |
| --- | --- | --- |
| Secondary trunk road and branch road | 1200 | 30 |
| arterial road | 1350 | 40 |
| expressway | 1500 | 60 |
| Nanjing Vangtze river bridge | 1800 | 60 |

### 5.2.4. Urban Multi-Mode Traffic Network Section Flow Distribution

The travel OD demand is 7000 people. The multi-mode traffic network impedance and OD demand are introduced into the traffic allocation model of urban multi-mode traffic network. The algorithm helps obtaining the traffic distribution result for urban multi-modal traffic near the P + R interchange hub and the bus-and-subway interchange hub. Figure 8 is the urban traffic network near the traffic interchange hub.

According to the data from the Tables 4 and 5 and the actual urban traffic network structure, the following is concluded:

(1)　23 people choose the regular bus mode. The proportion of conventional public transfer only accounts for 0.33%. Long commute travelers are less preferable with such mode;

(2)　Car travel enjoys the highest proportion of all travel modes, which occupies 66.89%. In long-distance commuting, travelers are relatively comfortable and preferable for cars;

(3)　The ratio of public transportation to car travel is about 3:7. Though there are relatively few expressways and trunk roads available for urban road networks, cars are still the main mode of commuting.

**Table 4.** Table of partial urban network section allocation results.

| Section Number | OD | Service Level | Flow (Person) | Section Number | OD | Service Level | Flow (Person) |
|---|---|---|---|---|---|---|---|
| 0 | [1,2] | 0 | 2568 | 18 | [7,8] | 0.54 | 647 |
| 1 | [1,3] | 0 | 744 | 19 | [7,11] | 1.04 | 1246 |
| 2 | [1,4] | 0 | 868 | 20 | [8,7] | 0.81 | 975 |
| 3 | [1,5] | 0 | 714 | 21 | [8,12] | 0.91 | 1096 |
| 4 | [1,35] | 0 | 233 | 22 | [9,10] | 0.66 | 788 |
| 5 | [1,30] | 0 | 1856 | 23 | [10,13] | 2.60 | 4682 |
| 6 | [1,39] | 0 | 17 | 24 | [10,30] | 0 | 212 |
| 7 | [2,3] | 0.57 | 681 | 25 | [11,10] | 1.95 | 2342 |
| 8 | [2,4] | 0.84 | 1007 | 26 | [12,11] | 0.91 | 1096 |
| 9 | [2,7] | 0.73 | 880 | 27 | [13,14] | 1.55 | 2325 |
| 10 | [3,8] | 1.19 | 1425 | 28 | [13,15] | 1.57 | 2357 |
| 11 | [4,5] | 0.45 | 543 | 55 | [30,31] | 0.28 | 2295 |
| 12 | [4,6] | 1.33 | 1802 | 61 | [35,36] | 0.26 | 233 |
| 13 | [5,4] | 0.39 | 469 | 62 | [36,37] | 0.25 | 227 |
| 14 | [5,9] | 0.66 | 788 | 63 | [36,38] | 0 | 6 |
| 15 | [6,7] | 0.78 | 940 | 64 | [37,30] | 0 | 227 |
| 16 | [6,10] | 1.31 | 1764 | 66 | [39,40] | 0.03 | 23 |
| 17 | [7,6] | 0.75 | 902 | | | | |

**Table 5.** Travel Mode Results.

| Travel Mode | Number of Trips (Person/Time) | Proportion of Travel |
|---|---|---|
| Car | 4682 | 66.89% |
| P + R | 212 | 3.03% |
| Regular Bus | 23 | 0.33% |
| Regular bus-and-subway | 227 | 3.24% |
| Subway | 1856 | 26.51% |

*5.3. Phrase Summary*

This chapter is based on stochastic user equilibrium assignment model and multimode traffic assignment model, and uses the distribution of MSWA algorithm model, while integrates the contents from above two chapters to implement urban multimodal transportation network capacity coordination analysis. Taken the actual example, the urban traffic network analytical method is adopted to evaluate the network capacity coordination, and finally by changing the OD trips quantitative case of transferring hub passenger transfer ability mainly includes the following points in the conclusion:

(1) For long-distance transportation commuting, travelers prefer to travel by car, with a travel rate of 66.89%. However, the preference for traveling by conventional bus is very low, with a travel rate of only 0.33%, which is more consistent with the actual travel situation.

## 6. Analysis of Passenger Flow Guidance Ability of Conventional Bus-and-Subway Transfer Hub

Under the long-distance commuter travel scenario, the bus-and-subway interchange hub will enhance the attractiveness of the local conventional bus network, which converts the bus passenger flow into the subway passenger flow, thus achieving the balance of the local urban transportation network.

It could be obtained that the influencing factors on the coordination of the bus-and-subway interchange hub in the local urban transportation network, include the transfer walking time, the transfer waiting time and the transfer penalty. Due to little changes in the waiting time in subway transfer, this section only studies the impact of the transfer time and punishment of the bus-and-subway interchange hub on the coordination capacity of the urban transportation network.

Based on the sensitivity analysis, the traffic network analyzed in the case study was selected as the calculation example. The urban commuter travel volumes in the morning rush hour are 7000 people. The virtual transfer road section of bus-and-subway transfer hub was road section [37,30]. Values of the transfer walking time and transfer penalty in the section impedance function are shown in the Table 6. The parameters of other sections' impedance and multi-mode traffic network allocation model are consistent with those of the case study.

**Table 6.** Bus-and-subway Property Change Table.

| Bus + Subway Transfer Variable | | | | | | | |
|---|---|---|---|---|---|---|---|
| **Transfer Time for Walking (Minutes)** | 0 | 2 | 4 | 6 | 8 | 10 | 12 |
| **Transfer Penalty (Minutes)** | 5 | 7.5 | 10 | 12.5 | 15 | 17.5 | 20 |

### 6.1. Analysis on the Impact of Transfer Walking Time

The transfer walking time of bus-and-subway transfer hub is taken as an influence variable into the urban multi-mode traffic network distribution model to obtain the distribution results of urban local traffic networks.

Table 7 Represents the change of transfer and walking time, and the change table of travel volume of each travel mode.

**Table 7.** Passenger flow scale of each travel mode.

| Transfer Walking Time (Min) | Travel Mode (Passenger Flow: Person) | | | | |
|---|---|---|---|---|---|
| | **Car** | **Subway** | **Regular Bus** | **P + R** | **Bus + Subway** |
| 0 | 4228 | 43 | 2044 | 245 | 439 |
| 2 | 4234 | 44 | 2097 | 252 | 373 |
| 4 | 4239 | 45 | 2144 | 257 | 315 |
| 6 | 4243 | 46 | 2184 | 262 | 265 |
| 8 | 4246 | 47 | 2220 | 266 | 222 |
| 10 | 4249 | 47 | 2249 | 269 | 186 |
| 12 | 4251 | 48 | 2275 | 272 | 154 |
| 14 | 4228 | 43 | 2044 | 245 | 439 |
| 16 | 4234 | 44 | 2097 | 252 | 373 |
| 18 | 4239 | 45 | 2144 | 257 | 315 |
| 20 | 4243 | 46 | 2184 | 262 | 265 |

Figure 9 shows the changes in the travel sharing rate of single-mode conventional bus travel, single-mode rail subway travel and the bus-and-subway transfer.

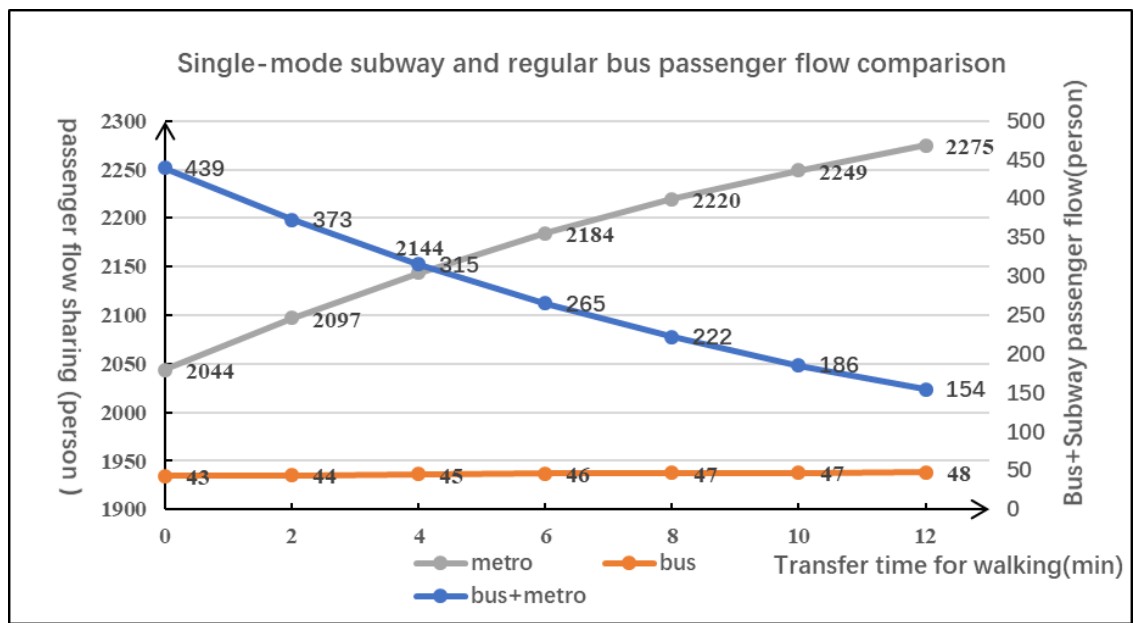

**Figure 9.** Comparison of Passenger Flow Between Single-mode subway and regular bus.

From the Table 7 and Figure 9 above, it could be concluded:

Among the urban traffic commuters, the proportion of car mode is the largest, while the proportion of conventional bus is the smallest. Changes of transfer walking time have no significant effect on the regular bus passenger flow;

With the increase of the transfer time, the change of the transfer volume of bus-and-subway shows a linear downward trend. When the transfer time is 0 min, the transfer volume is the highest (439 people). When the transfer time is 20 min, the transfer volume is the lowest (154 people);

Seen from the subway rail travel flow and the bus travel flow, with the increase of transfer waiting time, the bus-and-subway travel mode impedance enlarges. The bus-and-subway travel flow is mainly transformed into the subway travel flow. Travelers will choose the subway from the starting point of their travel. The local conventional bus network passenger flow decreases.

## 6.2. Analysis on the Impact of Transfer Penalty

The transfer penalty of conventional bus-and-subway transfer hubs is taken as an influence variable and involved into the urban multi-mode traffic network distribution model to obtain the distribution result of urban local traffic network. In addition, the bus in this case refers to the passenger flow of commuters who only use conventional buses to travel.

Table 8 represents the change of transfer punishment and the change of travel amount of each travel mode.

**Table 8.** Passenger flow scale of each travel mode.

| Transfer Walking Time (Min) | Travel Mode (Passenger Flow: Person) | | | | |
|---|---|---|---|---|---|
| | Car | Subway | Car | P + R | Bus + Subway |
| 5 | 4231 | 2072 | 44 | 248 | 405 |
| 7.5 | 4238 | 2133 | 45 | 256 | 329 |
| 10 | 4243 | 2185 | 46 | 262 | 265 |
| 12.5 | 4247 | 2227 | 47 | 267 | 212 |
| 15 | 4250 | 2263 | 48 | 271 | 169 |
| 17.5 | 4252 | 2291 | 48 | 274 | 134 |
| 20 | 4255 | 2314 | 49 | 277 | 106 |

The Figure 9 shows the changes of the travel sharing rate of single-mode conventional bus travel, single-mode subway travel and the bus-and-subway transfer.

Following conclusions can be drawn from the Table 8 and Figure 10:

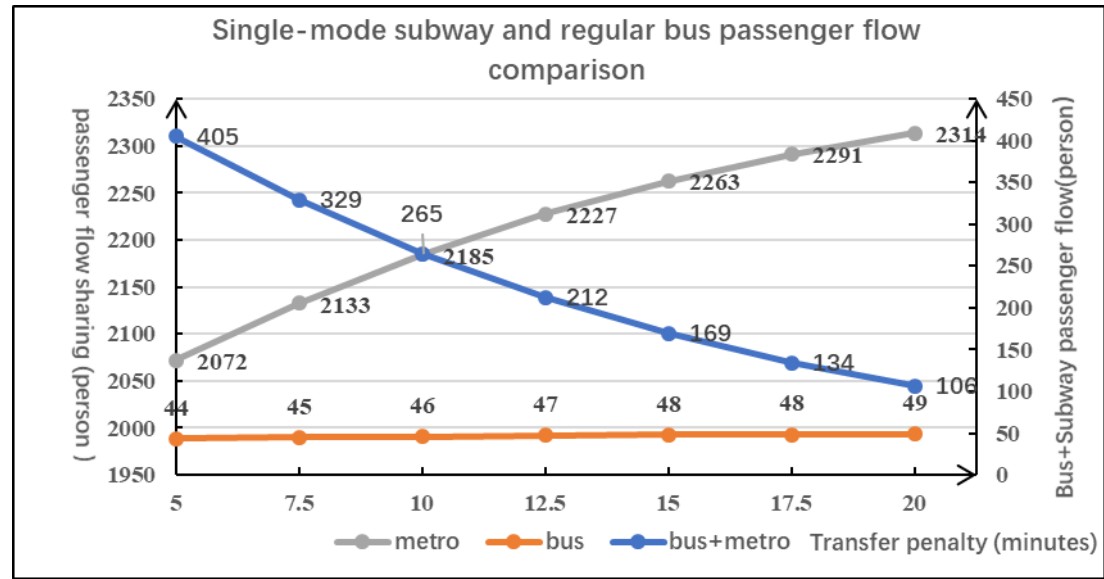

**Figure 10.** Comparison of Passenger Flow Between Single-mode Subway and Regular Bus.

Among the urban traffic commuters, the proportion of car mode is the highest while the proportion of conventional bus is the lowest. Changes of transfer penalty have no obvious effect on the regular bus passenger flow;

With the increase of transfer punishment, the transfer volume of bus-and-subway transfer presents a linear downward trend. When the transfer penalty is 0 min, the transfer volume is the highest (405 people). When the transfer penalty is 20 min, the transfer volume is the lowest (106 people);

It can be seen from the subway-and-bus transfer that, with the increase of transfer punishment, the impedance of bus-and-subway transfer mode increases. The bus-and-subway transfer mode is mainly transformed into the subway rail transfer. Travelers will choose subway mode from the starting point of their travel. Moreover, the passenger flow of local conventional bus network decreases.

## 7. Conclusions

Contemporarily, the urban traffic network congestion is intensified and highly concerned by the public. The urban traffic is being transformed from a single road network to a multi-mode traffic network which consists of the subway network, the conventional bus network, the road network and so on. Through the transportation interchange hub, those networks are connected to effectively alleviate local traffic congestion. When the service level of certain types of transportation networks is insufficient, they can pass through the interchange hub and transfer traffic demand to other modes of transportation networks to maximize the overall efficiency of the integrated transportation system.

Therefore, considering the multi-modal transportation network and traffic interchange hub composed of cars, conventional buses and subways, the influence of multi-modal traffic network on passenger travel behavior is analyzed. The impedance function of multi-mode traffic network is established. Urban multi-modal transportation network topology is constructed. The role of interchange hubs in coordinating multi-mode transportation network capacity is explored. Based on this, this paper has come up with following conclusions:

1.  In the case of long-distance commuting, the bus-subway transfer hub will enhance the attraction of the local regular bus network, transform the bus passenger flow into the subway passenger flow, and realize the balance of the local urban transportation network;

2. Under the long-distance vector travel scenario, the proportion of car mode is the largest, while the proportion of regular bus is the smallest. Changes of transfer walking time have no significant effect on the regular bus passenger flow;

3. Under the long-distance vector travel scenario, the proportion of car mode is the highest while the proportion of regular bus is the lowest. Changes of transfer penalty have no obvious effect on the regular bus passenger flow.

**Author Contributions:** The authors confirm contribution to the paper as follows: study conception and design: Y.Y. and J.C.; data collection: Y.Y. and Z.D.; analysis and interpretation of results: Y.Y. and Z.D.; draft manuscript preparation: Y.Y. All authors reviewed the results and approved the final version of the manuscript.

**Funding:** The authors acknowledge that the research is supported by the Key Project of National Natural Science Foundation of China (No. 51638004).

**Conflicts of Interest:** The authors declare no conflict of interest.

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
