# Peer review of "Analysis of the Passenger Flow Transfer Capacity of a Bus-Subway Transfer Hub in an Urban Multi-Mode Transportation Network"

_sustainability, doi:10.3390/su12062435_

Round 1

Reviewer 1 Report

The paper intends to integrate several predicting models to predict traffic flows in medium and large-size urban areas.

It would be better to give a clearer explanation about the choices of models, i.e. super-network model, random user equilibrium distribution model, MSWA algorithm and etc.

The conclusion section should provide more details and discussion about the analytical results from previous section.

Some detailed comments:

  1. A shortage of references; the length of the paper; the use of terms should be consistent;
  2.  
  3. Some claims need more evidences, otherwise they are assertions.
    • 3 Line 7-8 “Among the studies, the multi-mode traffic network model based on super network stands out”;
    • 4 Line 14-17;
    • 7 Line 5-9;
    • 7 Line 21, where did the argument “travelers' sense of comfort depends on the mode of transportation and travel time” come from?
    • 8 Line 12 Why was that “travel cost and comfort loss of the section are all calculated as zero”?
    • 8 Line 23 Why was that “traffic delay at road intersections is not considered”?
    • 9 Line 7 Why was that “the comfort loss function of a car is related to its travel time on the road”?
    • 9 Line 22-23 Whilst most of conventional buses do not run on the bus lanes, why did this research assume that buses ran on the bus lanes?
    • 9 Line 28-31 In reality, bus fares are considered according to many factors. For example, the fare is compared with income level to judge whether it is affordable; how the fare is charged in multiple mode public transport network (e.g. discount when transfer from bus to metro); some bus routes are charged by distance, and etc. The setting in this research is over-simplified.
    • 14 Line 13-14 What was the super network SUE model problem?
    • 19 Line 23-26;
    • 19 Line 29 “Due to little changes in the waiting time in subway transfer…” Quantify it.
  4. Solid lines in Figure 2, 3 and 4 could represent a variety of attributes. To what extent do these attributes interact?
  5. 7 Line 11, the definition of “large and medium-sized cities”?
  6. 8 Line 2, the definition of “upper network segment”? Line 11 “lower network segment”?
  7. 15 Line 15-19, the scope of the study area should provide basic information such as population and area (square kilometer) to give readers a general idea about the study area and reflect the claim that “this paper is based on the traffic travel in large and medium-sized cities”;
  8. Figure 7 should be able to provide sufficient information about the description at P.16 Line 1-17;
  9. 16 Line 28 unfinished sentence;
  10. 18-19 Table 4&5: the source of data? Or the result of computer simulation?
  11. 19 Table 5: The ratio of bus passengers was in doubt because it was hugely different from other cities around the world; more background information is needed to explain a) geographic distance between spots in Figure 7; b) the reason passengers did not choose bus services, was that because of poor bus quality, inaccurate bus schedules, insufficient bus vehicles, inadequate bus routes, or something else?
  12. 19 Line 1-20: no discussion about Subway; definition of “long-distance commuting”;
  13. 20 Table 7, was that Transfer “walking” time or “waiting” time? Contradictory to P.21 Line 6-7. Moreover, numbers in Table 7 and Figure 9 were contradictory;
  14. 22 Line 4, should be “Figure 10”;
  15. Unfinished conclusion section which did not present the main findings about this research, only the methods were described.

Reviewer 2 Report

The topic is very interesting and challenging. However it is necessary to have mode precise  explanation in the conclusion. The Figure 9 and 10 are very important message as the conclusion. However there are no detail explanation between the result of each graphs and equation (22) which is the core of this research. And there are several missing explanation and wrong parameter definition.

  1. there is no explanation of eta of equation (3)
  2.  L27P8 pcu/h is traffic volume, not capacity
  3. L8, L29 P10 unit vehicle/h is used but L27P8 is used pcu/h. need to be unified.
  4. L16P12 parameter I in equation (19)(29) have to be explained . what is parameter I unit (time) ?
  5. In table 4, what is "Traffic (person)" ? Does it mean "Trip" of person ?
  6. The travel efficiency in this research seems time based according to equation (22). It is necessary to explain how to calculate travel efficiency.
  7. In Figure 10, there is missing Usage Guide like Figure 9.

Please check the total research story and more additional explanation between conclusion Figure (9), (10) and equation which authors introduced in this paper. 

Round 2

Reviewer 1 Report

My questions have been answered.  There is no further question.  Having said that, English language and style requires proofreading.

Author Response

Response: I have revised my English in the revised draft.

Reviewer 2 Report

Dear, Author. Thank you for correcting and update your manuscript. It becomes much better than the first draft.  One suggestion is P26L9 item 2 should be new line for better looking.

Author Response

Response: I have corrected my reference format in the revised draft.